# Chitosan-Hydrogel Polymeric Scaffold Acts as an Independent Primary Inducer of Osteogenic Differentiation in Human Mesenchymal Stromal Cells

**DOI:** 10.3390/ma13163546

**Published:** 2020-08-11

**Authors:** Simona Bernardi, Federica Re, Katia Bosio, Kamol Dey, Camillo Almici, Michele Malagola, Pierangelo Guizzi, Luciana Sartore, Domenico Russo

**Affiliations:** 1Department of Clinical and Experimental Sciences, Bone Marrow Transplant Unit, ASST Spedali Civili, University of Brescia, 25123 Brescia, Italy; federicare91@gmail.com (F.R.); katia.bosio49@gmail.com (K.B.); michele.malagola@unibs.it (M.M.); domenico.russo@unibs.it (D.R.); 2Centro di Ricerca Emato-Oncologica AIL (CREA), ASST Spedali Civili, 25123 Brescia, Italy; 3Department of Mechanical and Industrial Engineering, University of Brescia, 25123 Brescia, Italy; k.dey@unibs.it (K.D.); luciana.sartore@unibs.it (L.S.); 4Laboratory for Stem Cells Manipulation and Cryopreservation, Department of Transfusion Medicine, ASST Spedali Civili, 25123 Brescia, Italy; camillo.almici@asst-spedalicivili.it; 5Orthopedics and Traumatology Unit, ASST-Spedali Civili, 25123 Brescia, Italy; pierangelo.guizzi@asst-spedalicivili.it

**Keywords:** osteopontin, scaffold, hydrogel, chitosan, regenerative medicine, osteogenic differentiation, digital PCR

## Abstract

Regenerative medicine aims to restore damaged tissues and mainly takes advantage of human mesenchymal stromal cells (hMSCs), either alone or combined with three-dimensional scaffolds. The scaffold is generally considered a support, and its contribution to hMSC proliferation and differentiation is unknown or poorly investigated. The aim of this study was to evaluate the capability of an innovative three-dimensional gelatin–chitosan hybrid hydrogel scaffold (HC) to activate the osteogenic differentiation process in hMSCs. We seeded hMSCs from adipose tissue (AT-hMSCs) and bone marrow (BM-hMSCs) in highly performing HC of varying chitosan content in the presence of growing medium (GM) or osteogenic medium (OM) combined with Fetal Bovine Serum (FBS) or human platelet lysate (hPL). We primarily evaluated the viability and the proliferation of AT-hMSCs and BM-hMSCs under different conditions. Then, in order to analyse the activation of osteogenic differentiation, the osteopontin (*OPN*) transcript was absolutely quantified at day 21 by digital PCR. *OPN* was expressed under all conditions, in both BM-hMSCs and AT-hMSCs. Cells seeded in HC cultured with OM+hPL presented the highest *OPN* transcript levels, as expected. Interestingly, both BM-hMSCs and AT-hMSCs cultured with GM+FBS expressed *OPN*. In particular, BM-hMSCs cultured with GM+FBS expressed more *OPN* than those cultured with GM+hPL and OM+FBS; AT-hMSCs cultured with GM+FBS presented a lower expression of *OPN* when compared with those cultured with GM+hPL, but no significant difference was detected when compared with AT-hMSCs cultured with OM+FBS. No *OPN* expression was detected in negative controls. These results show the capability of HC to primarily and independently activate osteogenic differentiation pathways in hMCSs. Therefore, these scaffolds may be considered no more as a simple support, rather than active players in the differentiative and regenerative process.

## 1. Introduction

Regenerative medicine, as well as bio-engineering, aims to repair or replace poorly functioning tissues or organs and holds promise in a wide range of fields and applications [1,2]. The key players of present regenerative medicine approaches are human mesenchymal stromal cells (hMSCs) and three-dimensional (3D) scaffolds [3], combined together.

hMSCs can be isolated from many tissues and organs such as bone marrow, dental pulp, adipose, synovium, and birth-derived tissues. However, hMSCs derived from the different tissue of origin present unique or varied levels of regeneration capabilities [4]. The promising evidence for hMSCs medical applications success is based on their features: the ease of isolation, the facility of expansion in culture, the differentiation capacity in different cell types, and the immunomodulating ability related to anti-inflammatory conditions. Moreover, they present anti-microbial capability, and migratory capacity to injury sites, particularly appealing for regenerative purposes [4]. All these aspects are monitorable and evaluable during hMSCs cultures for tissue engineering or regenerative medicine purposes [5]. Moreover, hMSCs’ safety profile in allogeneic transplants and the limited ethical rules concerning their use enhance their application in regenerative medicine [6]. All the biological, physiological, and therapeutic functions of hMSCs are influenced by tissue source, method and anatomical district of cell harvest, isolation, and medical application procedures [7,8]. Among the tissue of origin, bone marrow (BM) [9] and adipose tissue (AT) [10,11] are the most studied and prevalent in the clinical trials of hMSCs [12].

Recently, the attention of researchers has moved from cells to their three-dimensional (3D) support in culture: the scaffolds [13]. The scaffold-relied approach depends on the use of appropriate structures supporting living cell attachment, proliferation, differentiation, and subsequent formation of 3D tissue [3,14]. Ideally, scaffolds should be biomimetic, biodegradable, and appropriately mechanically strong. Moreover, they should have optimal micropores enabling vascularization and allowing metabolic needs to be met by the cells [15]. Biocompatiblility and non-immunogenicity are also desirable features. Scaffolds should also be versatile concerning manufacturing methods, functionalization potential, and control of three-dimensional microarchitecture [3,16]. It is important to understand that the bio-physiological and chemical 3D organization of cells is fundamental for a successful design and maturation of new tissues. Cells require a scaffold for structural and biological support. Scaffolds should guide cell growth and development toward their natural microenvironment, ensuring a balanced presence of specific biochemical (growth factors, differentiation signals, nutrients, and waste) and biophysical (pH and mechanical force regulation) cues to direct cell behavior [17,18].

In the last years, different data drove the research investigating the real role of the scaffold and polymers, in particular in osteogenesis and bone regeneration [19]. Are they simply supporting the cell cultures and the extracellular matrix (ECM) deposition, or are they active players in differentiation and in tissue development? Taherzadeh’s research group presented data concerning the increase in the osteogenic differentiation process promoted by scaffold fibronectin-coating when murine MSCs were equally stimulated by osteogenic medium [20]. Accordingly, morphological and molecular characterization of osteogenic differentiation performed on BM-hMSCs cultured in a 3D aragonite-based bi-phasic osteochondral scaffold confirmed the active role played by the scaffold in supporting osteogenic differentiation and enhancing cell proliferation [21]. Chimene and colleague demonstrated the osteoinductive capability of 3D bioprinted scaffolds on hMSCs, analysing both nascent ECM and transcripts in the absence of osteoinducing agents [22]. Similar results were previously obtained analysing titania containing phosphate-based glasses. The inorganic material was able to induce specific gene expression, alkaline phosphatase activation, and osteocalcin production [23].

Therefore, biochemical characteristics of the scaffolds seem to exert profound effects on cell fate.

One of the most investigated osteogenesis-related genes is osteopontin (*OPN*), together with alkaline phosphatase, *RUNX2*, and osteocalcin (*OCN*). OPN, also known as SPP1 (secreted phosphoprotein [1], is a secreted and chemokine-like glyco-phosphoprotein involved in the early phases of osteoblast differentiation and in other physiological and pathological processes, such as ECM mineralization, bone resorption, and bone tumor progression [24,25,26]. During osteoblast differentiation, several functional phases can be identified: proliferation, production, and maturation and mineralization of ECM [27]. *OPN* expression is very well controlled during physiological osteoblast differentiation and is not stable [28].

In this study, we focus in particular on a 3D hydrogel-chitosan scaffold (HC). In recent years, considerable attention has been given to chitosan-based scaffolds and their applications in the field of bone regeneration and orthopedic tissue engineering. Chitosan is a natural carbohydrate biopolymer derived from chitin, present in abundance in the exoskeletons of crustaceans and insects. Chitosan presents many features making it a good candidate for biomedical applications: it is biocompatible, biodegradable, bioactive, non-immunogenic, non-toxic, cost-effective, with antimicrobial capability, and shows haemostatic activity [29]. In addition, its structure mimicking natural ECM constituents and cationic surface charges is expected to promote cell attachment and growth [30]. Since chitosan has biological and chemical similarities to natural tissues, it has been shown that it is able to trigger the proliferation of fibroblasts, osteoblasts, and chondroblasts as well as the differentiation of osteoprogenitor cells and also to induce bone formation [31].

In this work, we investigated the capability of HC [32] to independently induce osteogenic differentiation in hMSCs cultured with growing medium (GM) or in osteogenic medium (OM) by *OPN* and *OCN* quantification. In order to evaluate the molecular adipogenic and chondrogenic differentiation pathway activation, we absolutely quantified Fatty Acid Binding Protein 4 (*FABP4*) and aggrecan, respectively. Commonly, *OPN* expression has been investigated by conventional real-time Polymerase Chain Reaction (PCR) or by transcriptome analysis. In this study, we quantified the *OPN* transcript by a highly sensitive and cost-effective digital PCR (dPCR) approach.

## 2. Results

### 2.1. HC1 and HC2 Scaffold Characterization

Two different three-dimensional scaffolds, namely, HC1 and HC2, were synthesized by means of a simple, cell-friendly synthesis method in aqueous condition without using any solvents, chemical reagents such as coupling agents, catalysts, and so on. A chemical crosslinking involving covalent bounds between the natural component G or CH and PEG was obtained, and by varying the CH concentration, two different hydrogels, namely, HC1 (8.1% by weight of CH) and HC2 (14.9% by weight of CH), were prepared (Table 1).

The hydrogel morphology was analyzed by a scanning electron microscope (SEM). Both hydrogels showed a micro-macro porous network with channels and interstices of different sizes, well interconnected and homogeneously distributed. In particular, large macroporous channels were visible in the side views of the hydrogels while irregular spherical pore morphologies were observed in the cross-sectional views, indicating the creation of anisotropic structures. These void channels were connected by large-area lamellae that were composed of both macro and micro pores ranging from 10 to about 450 μm. The distance between lamellae was some tens to more than 200 μm, and they were interconnected with tiny bridges. Data are shown in Figure 1.

The porosity was measured using the ethanol displacement method [33] and was found to be 78% and 81%, respectively, for HC1 and HC2.

The complete structural and mechanical characterization of the chitosan-based scaffold has been previously reported [32,34].

### 2.2. Viability and Cell Proliferation

hMSCs viability in the scaffolds was evaluated in three replicates using a Live/Dead assay after 21 days in complete medium FBS or human Platelet Lysate (hPL). BM-hMSCs and AT-hMSCs were viable and distributed homogenously at analysis time points, both in the 3D hydrogel-chitosan scaffold 1 (HC1 with 8.1% of chitosan) and in scaffold 2 (HC2 with 14.9% of chitosan). Specifically, cell viability within the hydrogels ranged between 75% and 90% for all conditions, independently of the medium composition, of the content of CH in the scaffold, and of cell type, as previously described [32]. Data are shown in Appendix A.

Cell proliferation was based on the number of viable cells within the scaffolds detected by the CCK8 assay at different time points. We observed that both BM-hMSCs and AT-hMSCs proliferated over time, reaching a maximum cell number after 21 days of culture.

In particular, BM-hMSCs presented a gradual proliferation during culture, even if some significant differences were observed when considering different culture conditions. Specifically, at 10 days (Scaffold HC1 GM+hPL vs. GM+FBS, *p* = 0.0066; Scaffold HC1 GM+hPL vs. Scaffold HC2 GM+FBS, *p* < 0.0001) and at 14 days (Scaffold HC1 GM+hPL vs. GM+FBS, *p* = 0.0014; Scaffold HC1 GM+FBS vs. Scaffold HC2 GM+hPL, *p* = 0.011; Scaffold HC1 GM+hPL vs. Scaffold HC2 GM+FBS, *p* < 0.0001; and Scaffold HC2 GM+hPL vs. Scaffold HC2 GM+FBS, *p* < 0.0001). No significant difference was observed at 21 day of culture (Figure 2A).

BM-hMSCs cultured with GM+FBS presented a proliferation exponential trend over time, in the presence of both HC1 and HC2. Conversely, BM-hMSCs cultured with GM+hPL presented a controlled proliferation trend over time in the presence of both scaffolds: between 14 and 21 days of culture, the proliferation reached the so-called plateau or stationary phase, excluding hyper-proliferation (Figure 2A).

AT-hMSCs cultured in GM+hPL also presented a gradual proliferation exponential trend before 10 days of culture, followed by a stationary phase at 14 days of culture and by a second exponential proliferation phase at 21 days (Figure 2B). Significant differences emerged when analysing different culture conditions at 10 days of culture (Scaffold HC1 GM+hPL vs. GM+FBS, *p* = 0.0455; Scaffold HC2 GM+hPL vs. Scaffold HC1 GM+FBS, *p* = 0.0089; Scaffold HC1 GM+hPL vs. Scaffold HC2 GM+FBS, *p* = 0.003; Scaffold HC2 GM+hPL vs. GM+FBS, *p* = 0.0008) (Figure 2B).

As observed for BM-hMSCs, AT-hMSCs cultured in GM+FBS proliferated in the presence of both scaffolds over time. Nevertheless, low proliferation levels were detected between 6 and 14 days of culture while a statistically significant increase in the proliferating number of cells was observed between 14 and 21 days of culture (14 day-Scaffold HC1 GM+FBS vs. 21 day-Scaffold HC1 GM+FBS, *p* = 0.0044; 14 day-Scaffold HC2 GM+FBS vs. 21 day Scaffold HC2 GM+FBS, *p* < 0.0001), similarly with an exponential proliferation phase (Figure 2B).

We observed heterogeneity of the proliferation under each condition at a single time point both in BM-hMSCs and AT-hMSCs. Nevertheless, the proliferation trends result superimposable in all samples cultured under the same conditions.

No statistically significant difference was observed between BM-hMSCs and AT-hMSCs at any time point.

Osteogenic differentiation was molecularly evaluated by dPCR quantification of the transcript of *OPN* and *OCN*, two markers for the early steps of differentiation of stromal cells in osteoblasts. *OPN* and *OCN* transcripts were quantified at 21 days of both 3D and 2D cell cultures. The absolute number of copies of *OPN*, *OCN*, and *GAPDH* transcripts was obtained by the dedicated cloud software AnalysisSuite (Thermo Fisher) and expressed as number of dots/reaction.

*GAPDH*, the reference gene, had between 8000 and 8500 number of dots/reaction in all the analyzed samples. Therefore, considering the robustness of the quantification and the previously reported feasibility of dPCR absolute quantification [35], only the marker transcript quantification was considered and statistically analyzed, and no normalization was performed.

BM-hMSCs cultured with OM+hPL presented the highest quantity of the *OPN* transcript in both scaffold HC1 and scaffold HC2 (Scaffold HC1 OM+hPL vs. GM+FBS, *p* = 0.0001; Scaffold HC1 OM+hPL vs. GM+hPL, *p* < 0.0001; scaffold HC1 OM+hPL vs. OM+FBS, *p* < 0.0001; scaffold HC2 OM+hPL vs. GM+hPL, *p* < 0.0001; scaffold HC2 OM+hPL vs. OM+FBS, *p* = 0.0003). Moreover, both scaffold HC1 and scaffold HC2 induced the transcription of *OPN* in BM-MSCs when cultured with GM+FBS without other differentiation stimuli (HC1 GM+FBS vs. Negative control CTR-, *p* < 0.0001; HC2 GM+FBS vs. CTR-, *p*< 0.0001). (Figure 3A).

Under three conditions, the *OPN* transcript level quantified by dPCR resulted in higher basal conditions (GM+FBS) than under expected osteogenic differentiation-inducing conditions (Scaffold HC1 GM+hPL vs. GM+FBS, *p* = 0.0008; Scaffold HC1 OM+FBS vs. GM+FBS, *p* = 0.0010; Scaffold HC2 GM+hPL vs. GM+FBS, *p* = 0.0073).

In Figure 3B, the proliferation trends of BM-hMSCs cultured in both scaffold HC1 and HC2 in the presence of GM+FBS and GM+hPL are reported. Briefly, BM-hMSCs cultured in scaffold HC1 with GM+FBS proliferated more slowly than BM-hMSCs cultured in scaffold HC1 with GM+hPL. Conversely, the latter presented a lower absolute quantity of the OPN transcript. As a matter of fact, BM-hMSCs cultured in scaffold HC2 with GM+FBS presented the highest level of proliferation at 21 days of culture, when *OPN* was quantified. BM-hMSCs cultured in scaffold HC2 with GM+hPL presented a downward trend of proliferation between 14 and 21 days of culture, contemporary with *OPN* transcription lower than under basal conditions (*p* = 0.0073).

AT-hMSCs cultured with OM+hPL presented the highest quantity of the *OPN* transcript in both scaffold HC1 and scaffold HC2 (Scaffold HC1 OM+hPL vs. GM+FBS, *p* = 0.0237; Scaffold HC1 OM+hPL vs. GM+hPL, *p* = 0.0314; Scaffold HC1 OM+hPL vs. OM+FBS, *p* = 0.0215; Scaffold HC2 OM+hPL vs. GM+FBS, *p* = 0.0005; Scaffold HC2 OM+hPL vs. GM+hPL, *p* = 0.0011; Scaffold HC2 OM+hPL vs. OM+FBS, *p* = 0.0002), similar to what was observed in BM-MSCs. Moreover, both scaffold HC1 and scaffold HC2 induced the transcription of *OPN* in AT-MSCs when cultured with GM+FBS, without other differentiation stimuli (HC1 GM+FBS vs. CTR-, *p* = 0.0007; HC2 GM+FBS vs. CTR-, *p* = 0.0101). (Figure 4A).

AT-hMSCs cultured in the presence of hPL, independently from the medium (GM or OM), presented higher *OPN* transcript levels than when cultured in the presence of FBS (Scaffold HC1 OM+FBS vs. GM+hPL, *p* = 0.0475; Scaffold HC1 GM+hPL vs. GM+FBS, *p* = 0.0454; Scaffold HC2 GM+hPL vs. GM+FBS, *p* = 0.0177; Scaffold HC2 OM+FBS vs. GM+hPL, *p* = 0.0019). (Figure 4A)

In Figure 4B, the proliferation trends of AT-hMSCs cultured in both scaffold HC1 and HC2 in the presence of GM+FBS and GM+hPL are reported. AT-hMSCs cultured in HC1 in the presence of GM+FBS presented a proliferation exponential phase contemporary with the expression of *OPN*. Conversely, AT-hMSCs hMSCs cultured in HC1 in the presence of GM+hPL presented a downward trend of proliferation at 14 days, followed by a second proliferation exponential phase concomitant with an increase in *OPN* expression. The *OPN* transcript level of AT-hMSCs hMSCs cultured in HC1 in the presence of GM+hPL at 21 days was higher than that of AT-hMSCs cultured in HC1 in the presence of GM+FBS. Similar results were observed in AT-hMSCs cultured in HC2. (Figure 4B).

BM-hMSCs cultured with OM+hPL presented the highest quantity of the *OCN* transcript in both scaffold HC1 and scaffold HC2 (Scaffold HC1 OM+hPL vs. GM+hPL, *p* = 0.0006; scaffold HC1 OM+hPL vs. OM+FBS; scaffold HC2 OM+hPL vs. GM+FBS, *p* = 0.0299; scaffold HC2 OM+hPL vs. GM+hPL, *p* = 0.0109; scaffold HC2 OM+hPL vs. OM+FBS, *p* = 0.009). Moreover, both scaffold HC1 and scaffold HC2 induced the transcription of *OCN* in BM-MSCs when cultured with GM+FBS without other differentiation stimuli (HC1 GM+FBS vs. CTR-, *p* = 0.0043; HC2 GM+FBS vs. CTR-, *p* = 0.006). (Figure 5A).

Under three conditions, the *OCN* transcript level quantified by dPCR resulted in higher basal conditions (GM+FBS) than under expected osteogenic differentiation-inducing conditions (Scaffold HC1 GM+hPL vs. GM+FBS, *p* = 0.0056; Scaffold HC1 OM+FBS vs. GM+FBS, *p* = 0.0075; Scaffold HC2 GM+hPL vs. GM+FBS, *p* = 0.0171; Scaffold HC2 OM+FBS vs. GM+FBS, *p* = 0.0162).

In Figure 5B, the proliferation trends of BM-hMSCs cultured in both scaffold HC1 and HC2 in the presence of GM+FBS and GM+hPL are reported. The evidence obtained by analysis of the absolute quantification of *OCN* and the proliferation trends were superimposable with those observed by *OPN* quantification.

AT-hMSCs cultured with OM+hPL presented the highest quantity of the *OCN* transcript in both scaffold HC1 and scaffold HC2 (Scaffold HC1 OM+hPL vs. GM+hPL, *p* = 0.0218; Scaffold HC1 OM+hPL vs. OM+FBS, *p* = 0.0421; Scaffold HC2 OM+hPL vs. GM+FBS, *p* = 0.022; Scaffold HC2 OM+hPL vs. OM+FBS, *p* = 0.041), similar to that observed in BM-MSCs. Moreover, both scaffold HC1 and scaffold HC2 induced the transcription of *OPN* in AT-MSCs when cultured with GM+FBS, without other differentiation stimuli (HC1 GM+FBS vs. CTR-, *p* < 0.0001; HC2 GM+FBS vs. CTR-, *p* < 0.0001). (Figure 6A).

AT-hMSCs cultured in the presence of hPL within HC2, independently from the medium (GM or OM), presented higher *OCN* transcript levels than when cultured in the presence of FBS, even if no statistical significance was observed. Conversely, hPL seems to not contribute to the increase in OCN mRNA in AT-hMSCs cultured in the presence of hPL within HC1 (Figure 6A).

In Figure 6B, the proliferation trends of AT-hMSCs cultured in both scaffold HC1 and HC2 in the presence of GM+FBS and GM+hPL are reported. AT-hMSCs cultured in HC1 in the presence of GM+FBS presented a proliferation exponential phase contemporary with the expression of *OPN*. Conversely, AT-hMSCs hMSCs cultured in HC1 in the presence of GM+hPL presented a downward trend of proliferation at 14 days, followed by a second proliferation exponential phase concomitant with a decrease in *OCN* expression. The *OCN* transcript level of AT-hMSCs hMSCs cultured in HC1 in the presence of GM+hPL at 21 days was lower than that of AT-hMSCs cultured in HC1 in the presence of GM+FBS. Different results were observed in AT-hMSCs cultured in HC2. They presented results similar to what was observed in *OPN* expression (Figure 6B vs. Figure 4B).

### 2.3. Molecular Chondrogenic and Adipogenic Differentiation

In order to evaluate the capability of HC to induce chondrogenic and adipogenic differentiation, *aggrecan* and *FABP4* transcripts were quantify by dPCR, respectively. This quantification was performed to test the specificity of HC in molecular osteogenic differentiation induction.

BM-hMSCs presented a statistically significant decrease of *aggrecan* in the presence of hPL, in both scaffolds (Scaffold HC1 OM+hPL vs. OM+FBS, *p* = 0.044; Scaffold HC1 GM+hPL vs. OM+FBS, *p* = 0.047; Scaffold HC2 OM+hPL vs. OM+FBS, *p* = 0.0315; Scaffold HC2 GM+hPL vs. OM+FBS, *p* = 0.0324; scaffold HC2 GM+hPL vs. GM+FBS, *p* = 0.046). A trend was present when comparing scaffold HC1 GM+hPL vs. GM+FBS, *p* = 0.056. No significant difference was observed comparing the basal condition (GM+FBS) and negative control. Data are reported in Appendix A.

On the other hand, no statistically significant difference was observed when comparing *aggrecan* expression in AT-hMSCs cultured under different conditions in HC1. Similar results were obtained when analyzing *aggrecan* expression in AT-hMSCs cultured in HC2, with the exception of GM+FBS vs. OM+hPL *p* = 0.0404. No significant difference was observed comparing the basal condition (GM+FBS) and the negative control. Data are reported in Appendix A.

BM-hMSCs presented no statistically significant difference in the expression of *FABP4*, considering different culture conditions and the two different types of scaffold. No significant difference was observed when comparing the basal condition (GM+FBS) and negative control. Data are reported in Appendix A.

Conversely, AT-hMSCs presented a statistical significant increase in *FABP4* transcription in the presence of hPL (GM+hPL and OM+hPL). This phenomenon was observed in AT-hMSC cultured both in HC1 and in HC2 (Scaffold HC1 GM+hPL vs. GM+FBS, *p* = 0.0114; Scaffold HC1 GM+hPL vs. OM+FBS, *p* = 0.0125; Scaffold HC1 OM+hPL vs. OM+FBS, *p* = 0.0338; Scaffold HC2 GM+hPL vs. GM+FBS, *p* = 0.0213; Scaffold HC2 OM+hPL vs. GM+FBS, *p* = 0.0243; Scaffold HC2 OM+hPL vs. OM+FBS, *p* = 0.0251; Scaffold HC2 GM+hPL vs. OM+FBS, *p* = 0.0225). No statistically significant difference was observed in the *FABP4* transcript level when comparing AT-hMSCs cultured in the presence of GM+FBS and negative controls. Data are reported in Appendix A.

## 3. Discussion

Scaffolds are three-dimensional porous structures playing a pivotal role in regenerative medicine and tissue engineering for the repair of tissue and organs [36,37]. Different materials, both synthetized and printed, are suitable for cell culture and sustaining the development of native tissue [38,39,40].

Chitosan is a biomimetic, biodegradable, biocompatible, non-immunogenic, and versatile polymer presenting appropriate mechanical strength and anti-bacterial properties [16]. It is also able to support growth factor activity [32,34]. For these reasons, chitosan is used in tissue engineering, production of artificial organs, biotechnologies, and for the regeneration of damaged tissues [41,42,43]. Different chitosan-based scaffolds have been described in the scientific literature and all of them seem to be able to support cell growth and differentiation, with a focus on bone and cartilage regeneration [44,45].

We previously described the combination of chitosan, hydrogel, and PEG in 3D scaffolds with different percentages of chitosan. In particular, two different scaffolds, one with 8.1% (HC1) and one with 14.9% (HC2) chitosan, were tested and both of them presented ideal chemical, physical, and mechanical features for application in bone regeneration [32]. Both scaffolds presented resistance to tension and the maintenance of structural integrity under biodegradation. These are key elements for the adhesion, vitality, and proliferation of cells, but also for growth supplements, differentiation signals, nutrients, waste transport, and cell-to-cell interaction.

HC1 and HC2 have been re-tested in the present study in order to evaluate their independent capability to induce osteogenic differentiation.

BM-hMSCs and AT-hMSCs were cultured in HC1 and HC2 in the presence of GM+FBS and GM+hPL in order to confirm the viability and the proliferation of stromal cells within the hydrogel-chitosan scaffolds. The previously reported data were confirmed, and BM-hMSCs and AT-hMSCs presented a viability between 75% and 90% under all conditions, independent of the variables (scaffolds, cell type, and medium). Similar results were reported in different studies, and this is a turning point for the application of a scaffold with a peculiar chemical composition [46,47,48]. In fact, the structure and the materials must not interact with the vital function of the cells in order to be suitable for regenerative medicine application [49].

Concerning proliferation, BM-hMSCs cultured in the presence of GM+FBS presented a proliferation exponential phase both in HC1 and HC2 during the 21 days of monitoring. On the other hand, BM-hMSCs cultured in the presence of GM-hPL seemed to have an early control on proliferation on both scaffolds: between 14 and 21 days, the proliferation trend reached a plateau. This is an important observation since non-controlled hyper proliferation may be excluded. Significant differences have been reported in BM-hMSCs cultured under different conditions at 10 and 14 days of cultures, as reported in Figure 2A. In particular, BM-hMSCs cultured in the presence of GM+hPL proliferated more than those cultured in the presence of GM+FBS in both HC1 and HC2. It is likely this evidence is due to a double proliferative stimulus: one induced by the interaction cell-scaffold and one mediated by hPL. hPL is known to inter-play in physiological processes, e.g., proliferation, in different cellular populations, including BM-hMSCs [50,51,52]. After 21 days, no statistically significant difference was detected. We hypothesise that BM-MSCs cultured in the presence of GM+hPL control the proliferation because they reached a status of equilibrium between the space and number of cells. Conversely, BM-MSCs cultured in the presence of GM+FBS might need more time in order to reach the same equilibrium. This result is probably due to the absence of the proliferative stimulus given by hPL. Therefore, the proliferation is induced only by the scaffold, as reported in other studies [53,54,55].

On the other hand, AT-hMSCs in the presence of GM-hPL presented a first exponential proliferation phase until 10 days of culture, followed by a downward trend at 14 days, independent of the scaffold. A second exponential proliferation phase was revealed at 21 days of culture. It is likely that AT-hMSCs are able to activate a control pathway in order to down-regulate the proliferation after 14 days when cultured with GM-hPL. Moreover, statistically significance differences were present after 10 days of culture and they most likely reflect the double proliferative stimulus given by the scaffold and hPL to AT-hMSCs cultured in the presence of GM-hPL, similar to that observed in BM-hMSCs. Conversely, AT-hMSCs cultured in the presence of GM-FBS presented the exponential proliferation phase starting at 14 days. This late activation is probably due to the presence of a single stimulus: the scaffold, as previously reported [56]. Analogous to what was observed in BM-hMSCs, no statistically significant difference was detected after 21 days.

The capability of stem cells to control their proliferation is a key element in order to hypothesize the application of these models to regenerative medicine clinical trials. Hydrogel-chitosan scaffolds are inducers of cell proliferation, but no hyper-proliferation or uncontrolled mechanism was revealed. The cultured stromal cells probably activate control pathways when reaching equilibrium, avoiding neoplastic transformation.

This work focused on the evaluation of molecular osteogenic differentiation activation by *OPN* transcript quantification by dPCR. To our knowledge, this is the first study presenting the application of dPCR in the quantification of *OPN* in a regenerative medicine model. Usually, the *OPN* transcript is quantified by real-time PCR [57]. The OPN protein is quantified by immunofluorescence or colorimetric approaches. In the present study, we opted for dPCR since it seems to be more sensitive and robust than real-time PCR, and a limited input of nucleic acid is required [58,59].

Both scaffolds induced osteogenic differentiation by increasing OPN expression in the absence of other differentiation factors. In particular, in three cases, OPN was more expressed under basal conditions (GM+FBS) than in the presence of osteogenic differentiation inducers, as previously reported. In fact, the combination of OM and hPL was described as able to activate and finalize osteogenic differentiation thanks to the mineralization of the extracellular matrix and of osteogenic phenotype characterization [32]. Moreover, no *OPN* transcription has been detected in CTR-. The capability of 3D hydrogel-chitosan scaffolds to activate the osteogenic differentiation process by *OPN* transcription is a new and unexpected result. Other scaffolds have been reported to activate *OPN* expression [60,61,62], even if complete osteogenic differentiation was not obtained, similar to what we observed. It is likely that additional differentiative inducers are needed in order to finalize the differentiation process that resulted in the presence of OM or hPL.

Moreover, different *OPN* expression levels seem to reflect different proliferative trends, as reported in Figure 3 and Figure 4. *OPN* transcript quantification was possible only at day 21 of culture, and it is difficult to hypothesise the exact moment of *OPN* expression activation. Nevertheless, considering the dynamic proliferative profiles, we are probably observing the alternative induction of proliferation, differentiation, and of their respective control pathways mediated by the scaffold, the medium, and hPL [24,28]. We cannot consider OPN as a marker of complete osteogenic differentiation, but rather a marker of osteogenic differentiation induction. OPN is an early osteogenic marker, and its expression is strictly controlled but pivotal for the expression of secondary late markers. Only after the activation of the pathways for osteogenic commitment is OPN down-regulated. This is likely the mechanism in GM-hPL and OM-FBS culture conditions for both cell lines [63]. The highest *OPN* expression revealed in cells cultured in HC in the presence of OM and hPL may be the result of synergic stimuli given by these three elements, while the absence of one of them seems to cause a late differentiation activation. Moreover, BM-hMSCs and AT-hMSCs seem to activate different kinetic responses to these stimuli and it probably reflects their different phenotypes and cell-surface marker expression [64,65]. Similar results have been observed when analyzing the *OCN* transcript level. As reported in Figure 5 and Figure 6, *OCN* expression is increased by the presence of HC both in BM-hMSC and AT-hMSCs. Moreover, the highest *OCN* transcript levels were quantified under the stimuli of the scaffold, the osteogenic medium, and the hPL [32]. These results confirm the evidence observed when analyzing the *OPN* transcript and the capability of the HC scaffold to actively induce osteogenic differentiation.

Conversely, neither HC1 nor HC2 is able to activate the molecular pathways for chondrogenic and the adipogenic differentiation. Chondrogenesis and adipogenesis seem to be influenced only by hPL, as it is deduced by the quantification of *aggrecan* and *FABP4*, shown in Appendix A, respectively. In particular, *aggrecan* is strongly down regulated in BM-hMSCs by the presence of hPL. Similar results were obtained by *aggrecan* transcript quantification in AT-hMSC cultured in OM, even if with low statistical significance. The negative impact of hPL on chondrogenic differentiation of hMSCs cultured in 3D Hydrogel-chitosan scaffolds was previously described [34] and it is probably due to the presence of cell receptor-binding competitors within hPL. On the other hand, the quantified adipogenic marker presents the opposite behavior to the different stimuli: statistically significant differences were observed only in AT-hMSCs, as expected, while no differences came to light by *FABP4* transcript quantification in BM-hMSCs. Moreover, FABP4 expression was enhanced by hPL, as previously described by Shansky et al. hPL confirmed its capability to molecularly activate adipogenic differentiation [66], but this was not supported by the HC scaffold since no difference was observed when comparing the basal condition with negative controls. These results were observed in both HC1 and HC2.

Together, this evidence highlights the capability of 3D hydrogel-chitosan scaffolds to independently induce osteogenic differentiation in the absence of other stimuli in in BM-hMSCs and AT-hMSCs. This is probably mediated by mechanical stimuli of adherent cells. Conversely, these types of hydrogel scaffolds seem to not actively contribute to chondrogenic and adipogenic differentiation. These results support the hypothesis of an active role played by some particular scaffolds instead of simpler structural supports for biodegradation.

## 4. Materials and Methods

### 4.1. Synthesis of Chitosan-Based Hydrogels HC1 and HC2

The scaffolds used in this study were produced using the following reagents: type A gelatin (G) (pharmaceutical grade, 280 bloom, viscosity 4.30 mPs), produced from pig skin, was a kind gift from Italgelatine, (Cuneo, Italy). Chitosan (CH) (molecular weight between 50,000 and 190,000 Da and degree of deacetylation 75–85%) was obtained from Fluka (Milan, Italy). Poly (ethylene glycol) diglycidyl ether (PEG) (molecular weight 526 Da) was supplied by Sigma–Aldrich (Milan, Italy). Ethylene diamine (EDA) and acetic acid were provided by Fluka (Milan, Italy). All materials were used without further purification.

The scaffolds were fabricated as previously described [32]. Briefly, the scaffolds were produced starting from gelatin (G), chitosan (CH), and poly (ethylene glycol) di-glycidyl ether (PEG). The hydrogel scaffolds were prepared in aqueous solution, and the synthetic procedure involved the reaction between gelatine/chitosan amino-groups and the epoxy groups of functionalized PEG. The ratio between gelatin and the crosslinking agent PEG was 4.2 and it was constant for both the hydrogels synthesized. Briefly, G (6 g) was completely dissolved in 60 mL distilled water at 40 °C under mild magnetic stirring, and functionalized PEG (1.4 g) was added dropwise into the mixture followed by the addition of an established amount of CH solution (2 wt% CH in acetic acid solution) and EDA (50 µL). We prepared two CH-based hydrogels, namely, HC1 (8.1 wt% CH content) and HC2 (14.9 wt% CH content), by varying the chitosan concentration. The reaction mixture was gently magnetically stirred at 40 °C for 20’ and finally, it was poured into a glass plate for gel formation. The gel was cut into rectangular bars, frozen by dipping into a liquid nitrogen bath and then lyophilized. Finally, in order to further increase the degree of grafting and crosslinking, the sponge-like dried hydrogels were placed in an oven under vacuum for 2 h at 45 °C. Table 1 shows the composition of chitosan-based HC1 and HC2.

Cubic samples (5 × 5 × 4 mm) of each hydrogel composition were cut in a dry state by a mechanical saw and packed into vacuumed-sealed polyethylene bags. Prior to biological tests, packed dry hydrogels were sterilized by gamma irradiation with Cobalt 60 gamma rays using 27–33 kGy following UNI EN ISO 11,137 (Sterilization of Health Care Products).

### 4.2. Human BM and AT Mesenchymal Stromal Cells 2D and 3D Culture

For the purpose of the study, commercial human multipotent mesenchymal stromal cells (hMSCs) were used. hMSCs were expanded from BM (BM-hMSCs) and from AT (AT-hMSCs) (PromoCell, Heidelberg, Germany) in the presence of a growth medium (GM), high glucose-based Dulbecco’s Modified Eagle Medium (DMEM) supplemented with 2% L-glutamine/penicillin-streptomycin/amphotericin B solution (stock solution, 200 mM l-glutamine, 10.000 U/mL penicillin, 10 mg/mL streptomycin, 250 μg/mL amphotericin B), 1 mM sodium pyruvate, and MEM Non Essential Amino Acids Solution 1X.

Cells were cultured under these conditions for 21 days, hereinafter CTR- served as negative controls for differentiation analysis by digital PCR.

BM-hMSCs and AT-hMSCs were added to 10% FBS (hereinafter complete medium FBS) or 5% hPL (hereinafter complete medium hPL) at 37 °C and 5% CO_2_ in an incubator. hPL was used because of its previously described interaction in bone differentiation [66]. All samples were analyzed in triplicate.

At 80% confluence, adherent BM-hMSCs and AT-hMSCs cultured with FBS or hPL were detached from the flask using trypsin, washed in PBS and neutralized with complete medium FBS, centrifuged at 1100 rpm for 5′, resuspended in the GM, and counted with a hemocytometer. We slowly seeded 36 × 10^3^ cells at passage 3 directly on two different types of 3D hydrogel-chitosan scaffolds: scaffold 1 (HC1) with 8,1% and scaffold 2 (HC2) with 14.9% of chitosan, as described above. Cell/scaffold constructs were then incubated at 4 °C for 1 h and 37 °C under 5% CO_2_ conditions for 1 h in a 24-well non-adherent plate, as previously described [32]. After this time, 1 mL of complete medium FBS or complete medium hPL was added to each well. The cell culture medium was changed twice a week. Each construct was analyzed on day 21 for cell viability. Cell proliferation was analyzed at 2, 6, 10, 14, and 21 days. Moreover, the cells cultured in GM served as differentiation controls.

### 4.3. BM-AT-hMSCs Cell Viability and Cell Proliferation Assay

A Live/Dead kit for mammalian cells (ThermoFisher, Waltham, MA, USA) was used to analyze cell viability. Briefly, the samples were washed with Dulbecco’s phosphate-buffered saline (DPBS) and incubated for 30–45 min at RT in DPBS with 2 μM of calcein AM and 4 μM of ethidium homodimer-1 (EthD-1). To counterstain nuclei, NucBlue^®^ Live reagent (2 drops/mL) was added to the cultures. Live (stained in green with calcein AM) and dead (stained in red with EthD-1) cells were analyzed using a Zeiss Observer Z1 fluorescence microscope. Several images were taken from three different replicates of each sample, and the percentage of viable cells (% total cell number–% dead cells) was then calculated for each condition using Image J.

A Cell Counting Kit-8 (CCK-8, Sigma–Aldrich, Saint Louis, MI, USA) was used in order to analyze cell proliferation on days 2, 6, 10, 14, and 21 of cell culture. Three replicates of each cell/scaffold construct were moved to a new cell culture plate at each time point and incubated with a fresh culture medium containing the CCK-8 reagent (ratio 1:10) at 37 °C for 2 h 30 min. Then, the absorbance of 100 μL of supernatant transferred to a new cell culture plate was measured at 450 nm with an Infinite 200 PRO plate reader (Tecan, Männedorf, Switzerland). Absorbance at 450 nm is proportional to the number of viable cells in each sample. A calibration curve was constructed to determine the relative viable cell number in the hydrogels.

### 4.4. BM-AT-hMSCs Osteogenic Differentiation in the 3D Scaffold

A cell suspension at a cellular density of 10^6^ cells/mL (36 × 10^3^ cells/scaffold) was added to the scaffolds for osteogenic differentiation under static conditions in 24-uncoated well plates. After 48 h, 1 mL of complete medium FBS or complete medium hPL was substituted with osteogenic differentiation medium (OM), consisting of complete medium FBS or complete medium hPL supplemented with 10^−7^ M dexamethasone, 25 µg/mL L-ascorbic acid, and 3 mM NaH_2_ PO_4_. All samples were analyzed in triplicate. Cell culture medium was changed twice a week. As anticipated, hMSCs cultured in the complete medium FBS or complete medium hPL without additional osteogenic factors served as differentiation controls to evaluate the osteogenic potential of the hydrogels alone.

After 21 days of culture, morphological analysis of osteogenic differentiation was performed as previously described, while RNA was extracted and analyzed in order to investigate the molecular activation of differentiation.

### 4.5. RNA Extraction and Retro-Transcription

RNA extraction was performed using an RNeasy^®^ MiniKit (Qiagen, Hilden, Germany) following manufacturer’s instructions. Input materials were ≤5 mg of cells cultured within scaffolds under all conditions. Briefly, 350 μL of RLT Buffer with β-mercaptoethanol (1:100) were added to the samples. Omogenation was performed with TissueRuptor^®^; samples were then centrifuged at 20.000 rpm for 3′. Samples were added to one volume of ethanol 70% and then moved into an RNeasy MinElute column and centrifuged at ≥8000× *g* for 15″. Elution waste was discharged, and 350 μL of RW1 Buffer was added to each column. Samples were centrifuged at ≥8000× *g* for 15″, and elution waste was discharged. Then, 80 μL of DNase I solution (70 μL RDD Buffer RDD + 10 μL DNase I stock) was added to the column filter, and samples were incubated at room temperature for 15′. Then, 350 μL of RW1 Buffer was added to each column, and samples were centrifuged at ≥8000× *g* for 15”. Elution waste was discharged, and 500 μL of RPE Buffer was added to each column. Samples were centrifuged at ≥8000× *g* for 15″ and elution waste was discharged. Immediately, samples were added to 500 μL of ethanol 80% and centrifuged at ≥8000× *g* for 2′, followed by additional centrifugation at 20,000 rpm for 5′. Lastly, 14 μL of RNase-free water was directly added to the column membrane. After 10′, samples were centrifuged at 20,000 rpm for 10′ and elutions containing RNA were collected. In order to quantify the samples, 1 μL of RNA was tested by a NanoDrop 2000 c (Thermo Fisher Scientific, Waltham, MA, USA) and stored at −80 °C until use.

Retro-transcription was performed using an RnaUsScript Reverse Transcriptase kit (LeGene Biosciences-Twin Helix, Rho, Italy) following the manufacturer’s instructions. Briefly, a mix containing 1 μL of dNTPs 10 mM, 1 μL of Random Primer 50 ng/μL, and 1 μg of RNA was prepared for each sample. The reaction mix was incubated at 65 °C for 5′ on a Veriti thermalcycler (Applied Biosystems, Foster City, CA, USA). At the end of the incubation, samples were immediately put on ice, and 4 μL of Buffer, 1 μL of DTT, 4.50 μL of water and 0.50 μL of Reverse Transcriptase enzyme were added. Every reaction mix was then incubated at 25 °C for 5′, 50 °C for 45′, 70 °C for 15′ on a Veriti thermalcycler. In order to quantify the samples, 1 μL of the obtained cDNA was tested by a NanoDrop 2000 c (Thermo Fisher Scientific) and stored at −20 °C until use.

### 4.6. Digital PCR Analysis

In order to absolutely quantify *OPN*, *OCN*, aggrecan, and the *FABP4* transcript, a QuantStudio 3D Digital PCR system (Thermo Fisher Scientific, Waltham, MA, USA) was used. Commercial assays were used to target *OPN* (TaqMan Hs00959010 FAM-labeled), *OCN* (Hs01587814_g1 FAM-labeled), aggrecan (Hs00153936_m1 FAM-labeled), *FABP4* (Hs01086177_m1 FAM-labeled), and *GAPDH* (TaqMan Hs03929097 VIC-labeled), which served as a control for RNA extraction and retro-transcription procedures. All the assay were produced by Thermo Fisher Scientific, Waltham, MA, USA.

A pre-reaction mix containing 8 μL of Mastermix, 0.8 μL of GAPDH assay, 0.8 μL of *OPN* assay, and 5.4 μL of water was prepared for each sample. A standard volume of 1 μL of cDNA of sample was added to each pre-reaction mix, referred to as the reaction mix.

For the absolute quantification of *OCN, aggrecan* and *FABP4*, a pre-reaction mix containing 8 μL of Mastermix, 0.8 μL of targeting FAM-labeled assay, and 6.2 μL of water was prepared for each sample. A standard volume of 1 μL of cDNA of sample was added to each pre-reaction mix, referred to as the reaction mix.

A volume of 15 µL of reaction mix was loaded on a QuantStudio 3D Digital PCR 20K Chip (Thermo Fisher Scientific, MA, USA), a sample-one chip. The loaded chips were thermocycled at 95 °C for 8 min, 35 cycles at 95 °C for 15 s, and at 60 °C for 1 min, with a final extension step at 60 °C for 2 min. Copies/µL reaction were retrieved with QuantStudio 3D AnalysisSuite Cloud Software, and results were reported as number of positive reactions (dots).

### 4.7. Statistical Analysis

For the statistical studies of the in vitro assessment of cell proliferation and osteogenic differentiation on the hydrogel-chitosan scaffolds, GraphPad Prism (Ver 7.0) was used. A *t*-test with Welch’s correction, one-way ANOVA with Tukey’s post hoc test and two-way ANOVA with the Bonferroni post hoc test were performed. Three replicates of each sample were used. Statistical significance was accepted at the probability level *p* < 0.05.

## Figures and Tables

**Figure 1 materials-13-03546-f001:**
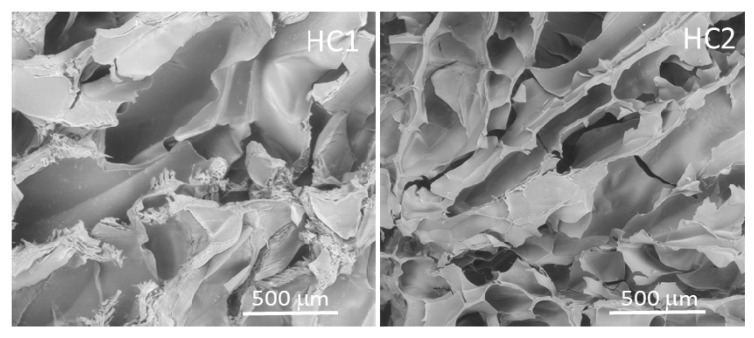
Scanning electron microscope images of chitosan-based HC1 and HC2 hydrogels.

**Figure 2 materials-13-03546-f002:**
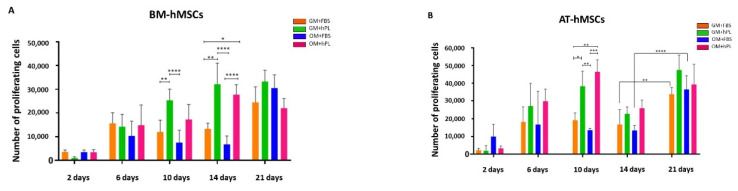
BM-hMSCs and AT-hMSCs proliferation in HC1 and HC2. (**A**) The figure reports the proliferation of BM-hMSCs cultured with GM+FBS or GM+hPL, in the presence of both HC1 and HC2, at different time points (2, 6, 10, 14 and 21 days). Data are expressed as the absolute number of proliferating cells in the sample. Statistical significances calculated by two-way ANOVA (* *p* ≤ 0.05, ** *p* ≤ 0.01, *** *p* ≤ 0.001, **** *p* ≤ 0.0001). (**B**) The figure reports the proliferation of AT-hMSCs cultured with GM+FBS or GM+hPL, in the presence of both HC1 and HC2, at different time points (2, 6, 10, 14 e 21 days). Data are expressed as the absolute number of proliferating cells in the sample. Statistical significances calculated by two-way ANOVA (* *p* ≤ 0.05, ** *p* ≤ 0.01, *** *p* ≤ 0.001, **** *p* ≤ 0.0001).

**Figure 3 materials-13-03546-f003:**
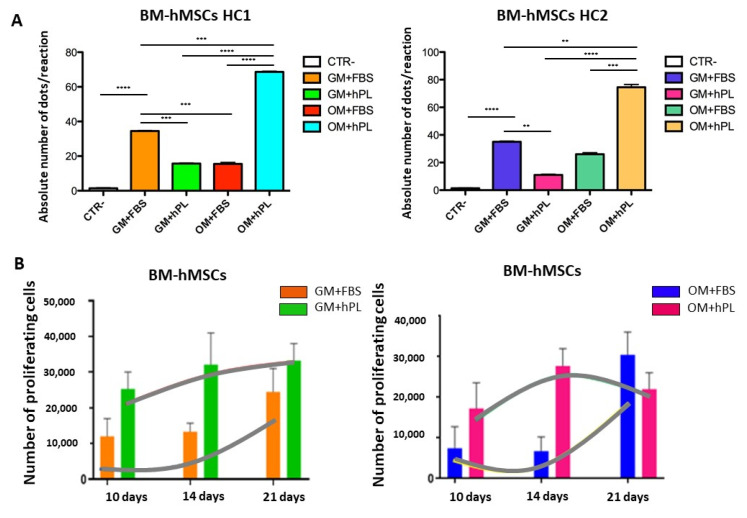
Osteogenic differentiation of BM-hMSCs evaluated by *OPN* transcript absolute quantification. (**A**) The figure reports the absolute quantification of the OPN transcript level obtained by dPCR analysis of BM-hMSCs at 21 days. All conditions are reported: GM+FBS, GM+hPL, OM+FBS, OM+hPL, for HC1. GM+FBS, GM+hPL, OM+FBS, OM+hPL, for HC2. CTR- obtained by 2D culture of BM-hMSCs cultured with GM served as a negative control. Data are expressed as the absolute number of positive dots/reaction. Statistical significances calculated by *t*-tests with Welch’s correction (* *p* ≤ 0.05, ** *p* ≤ 0.01, *** *p* ≤ 0.001, **** *p* ≤ 0.0001). (**B**) BM-hMSCs proliferation trends. Proliferating BM-hMSCs cultured in HC1 in the presence of GM+FBS and in the presence of GM+hPL, and cultured in HC2 in the presence of GM+FBS and in the presence of GM+hPL are reported at different time points (10, 14, and 21 days).

**Figure 4 materials-13-03546-f004:**
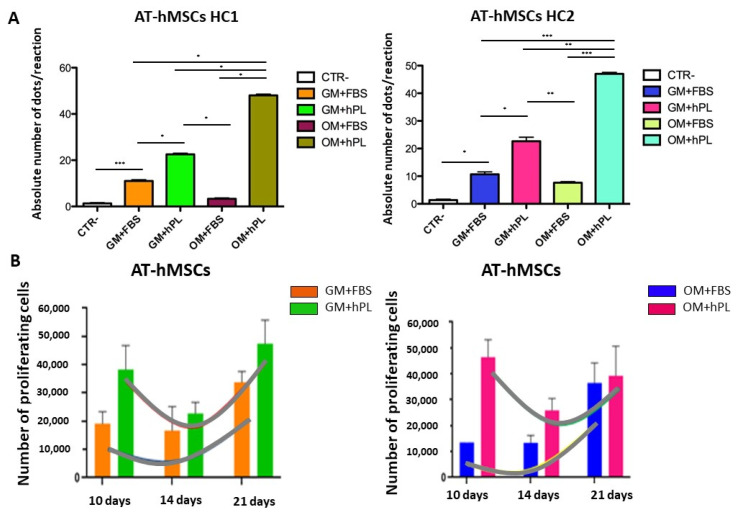
Osteogenic differentiation of AT-hMSCs evaluated by *OPN* transcript absolute quantification. (**A**) The figure reports the absolute quantification of the *OPN* transcript level obtained by dPCR analysis of BM-hMSCs at 21 days. All conditions are reported: GM+FBS, GM+hPL, OM+FBS, OM+hPL, for HC1. GM+FBS, GM+hPL, OM+FBS, OM+hPL, for HC2. CTR- obtained by 2D culture of BM-hMSCs cultured with GM served as a negative control. Data are expressed as the absolute number of positive dots/reaction. Statistical significances calculated by *t*-tests with Welch’s correction (* *p* ≤ 0.05, ** *p* ≤ 0.01, *** *p* ≤ 0.001, **** *p* ≤ 0.0001); (**B**) BM-hMSCs proliferation trends. Proliferating BM-hMSCs cultured in HC1 in the presence of GM+FBS and in the presence of GM+hPL, and cultured in HC2 in the presence of GM+FBS and in the presence of GM+hPL are reported at different time points (10, 14, and 21 days).

**Figure 5 materials-13-03546-f005:**
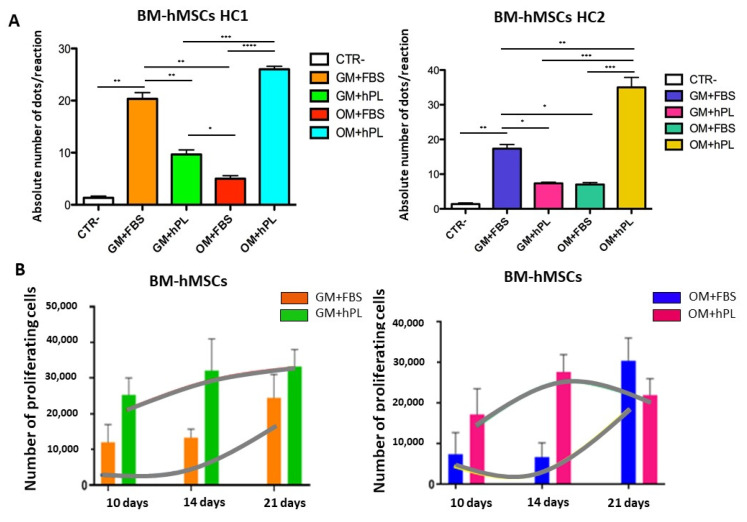
Osteogenic differentiation of BM-hMSCs evaluated by *OCN* transcript absolute quantification. (**A**) The figure reports the absolute quantification of the *OCN* transcript level obtained by dPCR analysis of BM-hMSCs at 21 days. All conditions are reported: GM+FBS, GM+hPL, OM+FBS, OM+hPL, for HC1. GM+FBS, GM+hPL, OM+FBS, OM+hPL, for HC2. CTR- obtained by 2D culture of BM-hMSCs cultured with GM served as a negative control. Data are expressed as the absolute number of positive dots/reaction. Statistical significances calculated by *t*-tests with Welch’s correction (* *p* ≤ 0.05, ** *p* ≤ 0.01, *** *p* ≤ 0.001, **** *p* ≤ 0.0001); (**B**) BM-hMSCs proliferation trends. Proliferating BM-hMSCs cultured in HC1 in the presence of GM+FBS and in the presence of GM+hPL, and cultured in HC2 in the presence of GM+FBS and in the presence of GM+hPL are reported at different time points (10, 14 and 21 days).

**Figure 6 materials-13-03546-f006:**
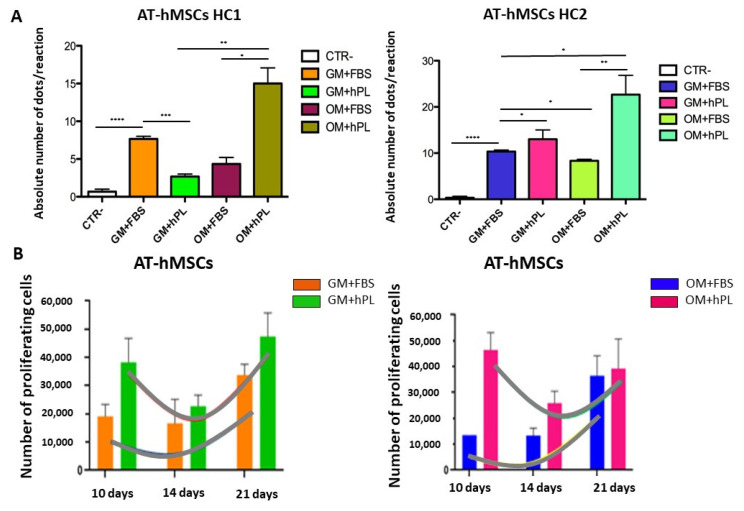
Osteogenic differentiation of AT-hMSCs evaluated by *OCN* transcript absolute quantification. (**A**) The figure reports the absolute quantification of the *OCN* transcript level obtained by dPCR analysis of BM-hMSCs at 21 days. All conditions are reported: GM+FBS, GM+hPL, OM+FBS, OM+hPL, for HC1. GM+FBS, GM+hPL, OM+FBS, OM+hPL, for HC2. CTR- obtained by 2D culture of BM-hMSCs cultured with GM served as a negative control. Data are expressed as the absolute number of positive dots/reaction. Statistical significances calculated by *t*-tests with Welch’s correction (* *p* ≤ 0.05, ** *p* ≤ 0.01, *** *p* ≤ 0.001, **** *p* ≤ 0.0001); (**B**) BM-hMSCs proliferation trends. Proliferating BM-hMSCs cultured in HC1 in the presence of GM+FBS and in the presence of GM+hPL, and cultured in HC2 in the presence of GM+FBS and in the presence of GM+hPL are reported at different time points (10, 14, and 21 days).

**Table 1 materials-13-03546-t001:** Composition and physical properties of the chitosan-based hydrogels.

	Composition			Physical properties		
Hydrogels	G (%)	PEG (%)	CH (%)	Gel Fraction (%)	Porosity (%)	Pore Sizes (µm)
HC1	74.3	17.6	8.1	84 ± 2	78 ± 3	10–450
HC2	68.8	16.3	14.9	82 ± 2	81 ± 7	10–450

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
