# Peer review of "Chitosan-Hydrogel Polymeric Scaffold Acts as an Independent Primary Inducer of Osteogenic Differentiation in Human Mesenchymal Stromal Cells"

_materials, 2020, doi:10.3390/ma13163546_

Round 1
Reviewer 1 Report
The authors Simona et al describe the osteogenic induction of gelatin/chitosan scaffolds independent of other factors such as media supplements. Two different scaffold variants are tested with either MSCs or ASCs with several different media formulations, with the primary measure of osteogenic induction being OPN expression.
- Figures should include legends to allow readers to more easily reference which colors refer to which experiment.
- The authors reference data on viability and proliferation in both the results and methods sections, however there is no viability data shown. The authors should include representative images used for the viability quantification. This will also show readers cellular morphology on the scaffolds.
- In the methods section, the authors reference previous publications regarding scaffold fabrication. A brief description of this process should be included in addition to the references.
- Osteogenic differentiation is the main focus of this manuscript with OPN expression measured via PCR being the only measure. The authors should include at least one other assessment such as histological stains for calcium using alizarin red or von Kossa’s. Additional genes assayed via PCR such as osteocalcin (OCN), RUNX2, Collagen 1 (COL1), and alkaline phosphatase (ALP) would also boost the significance of this manuscript.
- MSCs and ASCs have the capacity to preferentially differentiate in to osteoblasts, chondrocytes, and adipocytes. While ostengenic differentiation was strongest when directed down that lineage with OM-hPL media, differentiation (and OPN expression) in the GM-hPL and GM-FBS groups may be the result of spontaneous differentiation in the absence of any external factors. Expression of other transcripts should be assayed (via PCR) in all media conditions. Some suggested genes for chondrogenic differentiation are COL2A1 ,COL1A1, aggrecan, and adipogenic are FABP4, PPAR-gamma.
- Primer sequences should be listed in the methods section.
Author Response
Dear Reviewer, thank you so much for your kind revision and very important suggestions concerning the paper entitled “Chitosan-Hydrogel Polymeric scaffold acts as an independent primary inducer of osteogenic differentiation in human mesenchymal stromal cells” by Bernardi Simona et al.
Please, note that the page and line numbers reported referred to the manuscript version with tracked-changes.
Reviewer 1
The authors Simona et al describe the osteogenic induction of gelatin/chitosan scaffolds independent of other factors such as media supplements. Two different scaffold variants are tested with either MSCs or ASCs with several different media formulations, with the primary measure of osteogenic induction being OPN expression.
- Figures should include legends to allow readers to more easily reference which colors refer to which experiment.
Thank you for this kind suggestion. The legends have been added to all the figures. Please, see figures.
- The authors reference data on viability and proliferation in both the results and methods sections, however there is no viability data shown. The authors should include representative images used for the viability quantification. This will also show readers cellular morphology on the scaffolds.
We thanks the Reviewer for this point. We added the viability images in Supplementary materials, in Supplementary Figure 1. The figure was not included in the main text since the analysis focused only on proliferation and expression of osteogenic markers. Please, see Supplementary Materials, in particular Supplementary Figure 1.
- In the methods section, the authors reference previous publications regarding scaffold fabrication. A brief description of this process should be included in addition to the references.
Thank you for this kind suggestion. We added a paragraph concerning the scaffold fabrication. Please see “Materials and Methods” section, “Synthesis of Chitosan based Hydrogels HC1 and HC2” paragraph, pag 21 lines 27-45; pag 22 lines 1-8. Moreover, data concerning scaffolds characterization have been reported in “Results” section, in Figure 1 and in Table 2. Please, see pag 3 lines 30-45 and pag 4 lines 3-6.
- Osteogenic differentiation is the main focus of this manuscript with OPN expression measured via PCR being the only measure. The authors should include at least one other assessment such as histological stains for calcium using alizarin red or von Kossa’s. Additional genes assayed via PCR such as osteocalcin (OCN), RUNX2, Collagen 1 (COL1), and alkaline phosphatase (ALP) would also boost the significance of this manuscript.
We thanks the Reviewer for this suggestion. Considering that we are wondering the molecular activation of the osteogenic differentiation, we followed the Reviewer’s suggestion and we quantified by dPCR Osteocalcin. The test have been added both in “Materials and Method” and in “Results” section. In particular, the quantification of OCN confirmed the capability of HC to molecularly induce osteogenic differentiation. Please, see the related sections at pag 14-17, Figure 5 and Figure 6.
- MSCs and ASCs have the capacity to preferentially differentiate in to osteoblasts, chondrocytes, and adipocytes. While ostengenic differentiation was strongest when directed down that lineage with OM-hPL media, differentiation (and OPN expression) in the GM-hPL and GM-FBS groups may be the result of spontaneous differentiation in the absence of any external factors. Expression of other transcripts should be assayed (via PCR) in all media conditions. Some suggested genes for chondrogenic differentiation are COL2A1 ,COL1A1, aggrecan, and adipogenic are FABP4, PPAR-gamma.
We thanks the Reviewer for this important point that we did not consider. We quantified by dPCR aggrecan and FABP4. The analysis have been added both in “Materials and Method” and in “Results” section. Aggrecan resulted expressed both in control condition (CTR-) and in hMSCs cultured in presence of FBS, while aggrecan expression resulted inhibited by the presence of hPL. This phenomenon was previously described and we discussed it in the Discussion section. On the other hand, FABP4 resulted absent both in CTR- and in hMSCs cultured in FBS, while AT-hMSCs molecularly expressed this marker of adipogenic differentiation when culture in hPL. The capability of hPL to induce adipogenic differentiation in AT-hMSCs is known and we discussed it in the Discussion section. We did not observed the same results in BM-hMSCs. All the results are reported in Supplementary Figure 2 and Supplementary Figure 3, respectively. Please, see pag 18 lines 1-27; Pag 20 lines 48-50 and Pag. 21 lines 1-24 and the related Supplementary Figures 2 and 3.
- Primer sequences should be listed in the methods section.
We agree with the Reviewer that the primers and the probes sequences are very important, but we used commercial assays and the sequences are under patent. We reported all the assays numbers in “Materials and methods” section. Please, see pag. 24 lines 11-14.

Reviewer 2 Report
An interesting and striking ms of considerable interest to many.
To make a useful contribution to "Materials" the manuscript needs to contain a clear and complete description of the materials involved how the scaffolds were prepared and characterised. Since these are multicomponent scaffolds, it is important to understand how these components are distributed in the scaffold. From the ms statement of contributions this was the contribution of KD and LS
Minor points
P2 2nd para line 9. Include a ref to the important restrictions of the use of hMSCs
p4,p6,p8
Figures 1,2,3, The captions of these figures are very terse and addition words indicating the features in the figures which the authors may think is obvious to the TE reader but probably not in the broader scope of the Materials Journal.
I have marked this as major revision as the authors have not included information particularly relevant to their thesis "these scaffolds may be considered no more as simple support, rather as active players ….."
Author Response
Dear Reviewer, thank you so much for your kind revision and very important suggestions concerning the paper entitled “Chitosan-Hydrogel Polymeric scaffold acts as an independent primary inducer of osteogenic differentiation in human mesenchymal stromal cells” by Bernardi Simona et al.
Please, note that the page and line numbers reported referred to the manuscript version with tracked-changes.
Reviewer 2
An interesting and striking ms of considerable interest to many.
- To make a useful contribution to "Materials" the manuscript needs to contain a clear and complete description of the materials involved how the scaffolds were prepared and characterised. Since these are multicomponent scaffolds, it is important to understand how these components are distributed in the scaffold. From the ms statement of contributions this was the contribution of KD and LS.
Thank you for this kind suggestion. We added a paragraph concerning the scaffold fabrication. Please see “Materials and Methods” section, “Synthesis of Chitosan based Hydrogels HC1 and HC2” paragraph, pag 21 lines 27-45; pag 22 lines 1-8. Moreover, data concerning scaffolds characterization have been reported in “Results” section, in Figure 1 and in Table 2. Please, see pag 3 lines 30-45 and pag 4 lines 3-6.
Minor points
- P2 2nd para line 9. Include a ref to the important restrictions of the use of hMSCs
We really appreciated the suggestion of the Reviewer. We added the references number 6. Please, see Introduction section, pag 2 lines 16.
- p4,p6,p8 Figures 1,2,3, The captions of these figures are very terse and addition words indicating the features in the figures which the authors may think is obvious to the TE reader but probably not in the broader scope of the Materials Journal.
Thank you for this kind suggestion. In addition to the captions, the legends have been added to all the figures. Please, see figures.

Reviewer 3 Report
Dear authors,
The manuscript entitled "Chitosan-Hydrogel Polymeric scaffold acts as an independent primary inducer of osteogenic differentiation in human mesenchymal stromal cells" presents the differentiation of hMSCs into "osteocytes" utilizing specific growth media and chitosan-hydrogel polymer scaffolds. However major revisions must be performed, in order the quality of the manuscript to be improved.
1)In the introduction section, the authors should initially focus to the applications of chitosan-hydrogel polymer scaffolds in tissue engineering and regenerative medicine. Then can describe the potential use of hMSCs in these scaffolds. In this way, please describe in detail why the chitosan -hydrogel polymeric scaffolds are used in hard tissue engineering applications. Also, please change the paragraphs, accordingly in order to focus better to the main topic of the current manuscript.
2) The authors should declare if they used mesenchymal stem cells or multipotent mesenchymal stromal cells. Please take a look at the following references. a) Dominici M, Le Blanc K, Mueller I, Slaper-Cortenbach I, Marini F, Krause D, Deans R, Keating A, Prockop D, Horwitz E. Minimal criteria for defining multipotent mesenchymal stromal cells. The international society for cellular therapy position statement. Cytotherapy 2006; 8: 315-317 [PMID: 18397751 DOI: 10.1080/14653240600855905] and b)
Viswanathan S, Shi Y, Galipeau J, Krampera M, Leblanc K, Martin I, Nolta J, Phinney DG, Sensebe L. Mesenchymal stem versus stromal cells: International society for cell & gene therapy (isct(r)) mesenchymal stromal cell committee position statement on nomenclature. Cytotherapy 2019; 21: 1019-1024 [PMID: 31526643 DOI: 10.1016/j.jcyt.2019.08.002]
3) In the results, section Viability and cell proliferation, Figure 1, it would be better the diagramms of figure 1 to be combined into a new one. Seperately, can be used as supplementary data. Combining these two diagramms, it would be easier for the readers to undestand better the differences in proliferation between BM-hMSCs and AT-hMSCs. Also, perform statistical analysis between these two cell populations, in order to present differences and similarites.
4) In Figure 1, some of the presented bars have too high stadard deviation. What is the cause of this phenomenon, according to the authors. Maybe, if the authors increase the iniitial number of samples, the standard deviation to be reduced.
5) In Materials and Methods section, the authors should describe in detail the production of the chitosan-hydrogel polymer scaffods and their characteristics (size, pore size, biomechanical data if applicable).
6) In Materials and Methods section, the authors performed BM-AT hMSCs cell viability and cell proliferation assay. In the beginning please provide the immunofluoresence images of the live/dead assay. Also, what was the maximum depth that images were taken in oder to validate their viability. Typically, a confocal microscope can take images of about 10-100 μm from the surface. How do you know that these are representable images (Live/Dead cell assay).
7)In the same section, i strongly recomend to perform histological analysis to the recellularized scaffolds and to perform initially hematoxylin and eosin (H&E), alcian blue (AB) and von Kossa stains. These images must be represented in results section.
8)In Materials and Methods section. RNA extraction and Digital PCR analysis, the authors must confirm their results with a proteomic approach such as ELISA, wesern blod or equavalent or better proteomic approach.
9) In materials and Methods section, the authors should provide details regarding the expansion and proliferation of hMSCs. Moreover the minimum criteria (1 fibroblastic shape, differentiation to "ostetocytes", "adypoctes" and "chondrocytes", flow cytometric analysis). of ISCT must be fullfilled in order to conclude that these are true MSCs. Please provide all these details as supplementary files.
10) Please include all the information gathered from the results to discussion section.
Yours sincerely
Author Response
Dear Reviewers, thank you so much for your kind revision and very important suggestions concerning the paper entitled “Chitosan-Hydrogel Polymeric scaffold acts as an independent primary inducer of osteogenic differentiation in human mesenchymal stromal cells” by Bernardi Simona et al.
Please, note that the page and line numbers reported referred to the manuscript version with tracked-changes.
Reviewer 3
The manuscript entitled "Chitosan-Hydrogel Polymeric scaffold acts as an independent primary inducer of osteogenic differentiation in human mesenchymal stromal cells" presents the differentiation of hMSCs into "osteocytes" utilizing specific growth media and chitosan-hydrogel polymer scaffolds. However major revisions must be performed, in order the quality of the manuscript to be improved.
1) In the introduction section, the authors should initially focus to the applications of chitosan-hydrogel polymer scaffolds in tissue engineering and regenerative medicine. Then can describe the potential use of hMSCs in these scaffolds. In this way, please describe in detail why the chitosan -hydrogel polymeric scaffolds are used in hard tissue engineering applications. Also, please change the paragraphs, accordingly in order to focus better to the main topic of the current manuscript.
We really appreciate the suggestion of the Reviewer. The Introduction section was emended accordingly. We added some details concerning why the chitosan-hydrogel polymeric scaffolds are used in hard tissue engineering applications. Please, see “Introduction” section, pag 3 lines 10-20.
2) The authors should declare if they used mesenchymal stem cells or multipotent mesenchymal stromal cells. Please take a look at the following references. a) Dominici M, Le Blanc K, Mueller I, Slaper-Cortenbach I, Marini F, Krause D, Deans R, Keating A, Prockop D, Horwitz E. Minimal criteria for defining multipotent mesenchymal stromal cells. The international society for cellular therapy position statement. Cytotherapy 2006; 8: 315-317 [PMID: 18397751 DOI: 10.1080/14653240600855905] and b)Viswanathan S, Shi Y, Galipeau J, Krampera M, Leblanc K, Martin I, Nolta J, Phinney DG, Sensebe L. Mesenchymal stem versus stromal cells: International society for cell & gene therapy (isct(r)) mesenchymal stromal cell committee position statement on nomenclature. Cytotherapy 2019; 21: 1019-1024 [PMID: 31526643 DOI: 10.1016/j.jcyt.2019.08.002]
Thank you so much for this point. We used multipotent mesenchymal stromal cells. We added the details concerning the cells, since we used commercial cell lines. Moreover, all the text has been emended accordingly to this Reviewer’s suggestion. Please, see “Materials and methods” section, “Human BM and AT mesenchymal stem cells 2D and 3D culture” paragraph. Please, see pag 22 lines 11-12
3) In the results, section Viability and cell proliferation, Figure 1, it would be better the diagrams of figure 1 to be combined into a new one. Seperately, can be used as supplementary data. Combining these two diagrams, it would be easier for the readers to understand better the differences in proliferation between BM-hMSCs and AT-hMSCs. Also, perform statistical analysis between these two cell populations, in order to present differences and similarities.
We thank the Reviewer for this suggestion. We performed the statistical analysis between the two cell populations, but no statistical differences was observed. For this reason, considering that no comparison between the two cell lines has been reported, in order to not complicate the diagram, we opted to maintain the Figure 1, now namely Figure 2, split in Fig2A and Fig2B. We added the comment concerning the statistical analysis between BM-hMSCs and AT-hMSCs in the Results section. Please, see pag 5 lines 31-32.
4) In Figure 1, some of the presented bars have too high standard deviation. What is the cause of this phenomenon, according to the authors. Maybe, if the authors increase the initial number of samples, the standard deviation to be reduced.
We really appreciated this comment. We agree that the standard deviations are high in many conditions both for BM-hMSCs and for AT-hMSCs. We justify this phenomenon by the fact that it represents the heterogeneity of the proliferation in each condition at the single time point, while the proliferation trend of all the samples cultured in the same condition results superimposable. We added this comment in the “Results” section. Please, see pag 5 lines 28-30.
5) In Materials and Methods section, the authors should describe in detail the production of the chitosan-hydrogel polymer scaffolds and their characteristics (size, pore size, biomechanical data if applicable).
Thank you for this kind suggestion. We added a paragraph concerning the scaffold fabrication. Please see “Materials and Methods” section, “Synthesis of Chitosan based Hydrogels HC1 and HC2” paragraph, pag 21 lines 27-45; pag 22 lines 1-8. Moreover, data concerning scaffolds characterization have been reported in “Results” section, in Figure 1 and in Table 2. Please, see pag 3 lines 30-45 and pag 4 lines 3-6.
6) In Materials and Methods section, the authors performed BM-AT hMSCs cell viability and cell proliferation assay. In the beginning please provide the immunofluoresence images of the live/dead assay. Also, what was the maximum depth that images were taken in oder to validate their viability. Typically, a confocal microscope can take images of about 10-100 μm from the surface. How do you know that these are representable images (Live/Dead cell assay).
We thanks the Reviewer for this point. We added the viability images in Supplementary materials, since the analysis focused only on proliferation and expression of osteogenic markers. As reported in the Supplementary material caption, the images were taken at 5x, with a scale-bar of 100 μm. In particular, several images of different area of each scaffold-cells construct were snapped and analyzed. In this way, a complete analysis of the samples was performed. Please, see Supplementary Figure 1.
7) In the same section, i strongly recomend to perform histological analysis to the recellularized scaffolds and to perform initially hematoxylin and eosin (H&E), alcian blue (AB) and von Kossa stains. These images must be represented in results section.
We really appreciated this suggestion. We included the comment to this point together with one of point number 8.
8) In Materials and Methods section. RNA extraction and Digital PCR analysis, the authors must confirm their results with a proteomic approach such as ELISA, wesern blod or equavalent or better proteomic approach.
We thanks the Reviewer for this suggestion. The aim of this study was to test the capability of the scaffold to molecularly activate the osteogenic differentiation. We wondered the transcriptional activation of differentiation markers performed by the mechanical and physical forces of HC scaffolds on cells membrane. For this reason, all the samples were used for nucleic acids extraction. No proteic samples are available, since we were not evaluating the complete differentiation. For this reason we performed only molecular quantification. In order to confirm our results, we added the molecular quantification of OCN, aggrecan and FABP4 as markers of osteogenic, chondrogenic and adipogenic differentiation, respectively. Please, see Results section, Figure 5, 6, and Supplementary Figure 2 and 3. As reported in the Discussion, additional investigations are needed in order to evaluate the effects of the HC scaffolds on the differentiation of the cells during longer periods of culture and including also the histological analysis of the cells/scaffold structures in the different conditions. Please, see Results section, pag 14-17; Discussion section. Pag 20-21; Supplementary Figure 2 and Supplementary Figure 3.
9) In materials and Methods section, the authors should provide details regarding the expansion and proliferation of hMSCs. Moreover the minimum criteria (1 fibroblastic shape, differentiation to "ostetocytes", "adypoctes" and "chondrocytes", flow cytometric analysis). of ISCT must be fullfilled in order to conclude that these are true MSCs. Please provide all these details as supplementary files.
Thank you so much for this point. We used multipotent mesenchymal stromal cells. We added the details concerning the cells, since we used commercial cell lines. Moreover, all the text has been emended accordingly to this Reviewer’s suggestion. Please, see “Materials and methods” section, “Human BM and AT mesenchymal stem cells 2D and 3D culture” paragraph. Please, see pag 22 lines 11-12
10) Please include all the information gathered from the results to discussion section.
All the suggested comments and results have been added in the “Materials and Methods”, “Results” and “Discussion” sections.
Round 2
Reviewer 1 Report
The authors have made significant additions to the manuscript. Every comment made has been addressed, strengthening both the scientific merit, impact, and relevance to readers.
Included revisions relevant to this reviewer:
- Description of scaffold fabrications are now included in the Methods section.
- Live/Dead images are included in the supplementary data.
- Figure legends are included.
- Additional PCR experiments into OCN, Aggrecan, and FABP4 expression have been performed. Additional discussion pertaining to these results has also been included.
Reviewer 2 Report
The authors have responded well to the comments of the reviewers and have explained where the data requested was not available. The ms is now suitable for publication
Reviewer 3 Report
Dear Authors,
Thank you for performing the proposed revisions. In my opinion, I think that the quality of paper has been significantly improve and can be proceeded to the next step to the publication process.